# OpenReview forum: "DARC: Disagreement-Aware Alignment via Risk-Constrained Decoding"
_ICML.cc/2026/Conference — ICML 2026 regular_

### Official Review · Reviewer_CD6R · 2026-02-21

**Soundness:** 3
**Presentation:** 3
**Significance:** 3
**Originality:** 3
**Overall Recommendation:** 4
**Confidence:** 3

**Summary:**

This paper tackles the challenge of inference-time alignment for Large Language Models (LLMs) under systematic preference heterogeneity. Moving away from the standard practice of collapsing diverse human feedback into a single scalar reward, the authors elegantly reframe response selection as a risk-sensitive decision problem over satisfaction distributions.
To address this, the authors introduce DARC (Disagreement-Aware Alignment via Risk-Constrained Decoding), a retraining-free reranking framework. DARC relies on multiple preference samples (or disagreement proxies) to estimate the satisfaction distribution for candidate responses, selecting outputs by maximizing a KL-robust entropic value. A practical strength of the method is its deployment-time risk controls, which explicitly regulate the trade-off between mean satisfaction and robustness via an entropic risk premium. This is complemented by an $\epsilon$-tie-breaking mechanism designed to favor less controversial candidates among near-optimal options.
The theoretical grounding is solid: the authors derive uniform lower confidence bounds for expected satisfaction, successfully connecting their pessimistic selection rules to Distributionally Robust Optimization (DRO) and providing a clear justification for the KL-DRO entropic objective. They further generalize DARC for multi-scorer settings using a soft worst-case operator. Empirically, both automated and multi-annotator human evaluations confirm that DARC effectively improves tail robustness and mitigates disagreement-related risks without sacrificing average quality, with the most pronounced gains observed on high-disagreement prompts.

**Compliance With Llm Reviewing Policy:**

Affirmed.

**Final Justification:**

I have considered both the paper and the authors’ rebuttal, and I find that my main concerns have been sufficiently addressed. I therefore raise my score to Weak Accept.

The rebuttal improved my assessment of the paper’s soundness. In particular, my concern about the memory overhead of perturbation-based decoding was addressed with concrete VRAM measurements, which clarified the practical cost of the method. The response on proxy false positives was also useful, as it provided both mitigation heuristics and additional empirical results, making this aspect of the method clearer.In terms of originality, I still find the paper to contain a meaningful idea: framing decoding-time alignment as risk-constrained decision making under heterogeneous preferences is conceptually interesting, and the connection to robust entropic decoding gives the method a more principled basis.For significance, the rebuttal increased my confidence that the method may generalize beyond the main benchmark setting. In particular, the added evidence on hyperparameter transferability across math and code tasks addressed an important concern about robustness across domains.The rebuttal also improved clarity. The planned revisions, especially a clearer explanation of when to use different DARC variants, should make the paper easier to follow.

**Key Questions For Authors:**

1. **Memory overhead of perturbations:** Table 7 clearly demonstrates that the time latency overhead for generating perturbations is well under 3.2%. However, the paper does not address the potential VRAM bottleneck associated with decoding $N_{aug}$ sequences in parallel. Could you clarify the exact memory footprint required for these parallel rollouts, especially when scaling this approach to the 14B model evaluated in Appendix H.12?

2. **Mitigating proxy False Positives:** Appendix H.11 provides an excellent and honest taxonomy of proxy mismatch cases. Given that False Positives are heavily driven by the reward model's sensitivity to superficial surface-form changes, are there straightforward heuristics or calibration steps that could be applied during the perturbation phase to filter out these formatting artifacts?

3. **Hyperparameter transferability:** The ablation studies in Figure 2 effectively map out the performance impact of $\beta$, $\epsilon$, and $K$ on a fixed candidate pool. Have you tested how stable these optimal values are when transferring the method across fundamentally different domains (e.g., from open-ended conversational formats like MT-Bench to strictly objective coding or math benchmarks)?

**Limitations:**

yes

**Strengths And Weaknesses:**

## Strengths

* **Practical Significance:** The paper targets a critical failure mode in standard RLHF pipelines—the brittleness of mean-reward reranking under diverse human preferences. Offering a retraining-free, drop-in decoding rule is highly appealing for practical deployment.
* **Solid Theoretical Grounding:** The connection drawn between the KL-robust entropic objective and pessimistic LCB/DRO frameworks is neat. While the individual components (risk-sensitive objectives, soft worst-case aggregation) are established, synthesizing them into an explicit, risk-budgeted decoding rule for alignment is a solid and original contribution.
* **Rigorous Empirical Setup:** The authors carefully control for generation confounds by using fixed candidate pools. Reporting both mean and tail-oriented metrics provides a comprehensive view of the method's effectiveness.
* **Clear Narrative:** The overall structure of the paper is logical and easy to follow, transitioning smoothly from problem formulation to theory, algorithms, and finally evaluation.

## Weaknesses
* **Hyperparameter Clutter:** The method introduces a heavy load of hyperparameters ($\beta$, $\tau$, $\epsilon$) and closely related variants (constrained vs. penalized, multi-scorer). The paper currently lacks a sharp, practical "when-to-use-which" guide, making the framework feel overly complex to tune for practitioners.
* **Presentation and Formatting.** While the overall narrative is clear, there is a noticeable error in the figure sequencing. Figure 3 is introduced and placed on page 7 , whereas Figure 2 appears and is referenced later on page 8. The authors should swap the numbering of these figures so they appear in sequential order as they are introduced in the text.

---

> ### Author Rebuttal · Authors · 2026-03-30
>
> We thank the reviewer for the rigorous and insightful assessment, which helped significantly strengthen the paper’s empirical grounding.
> ## W1: Hyperparameter Organization & The "When-to-use-which" Guide
> We agree the paper would benefit from a clearer, more comprehensive practitioner-facing recipe. To address this, we will add a dedicated **“Variant Selection Guide”** in the revision to clarify the mechanics and intended use cases of each formulation:
> - **DARC-$\epsilon$ (default drop-in rule).** Our recommended general-purpose variant. Rather than imposing a hard penalty, it acts as a robust tie-breaker: it stays close to mean-based selection when disagreement is low, while falling back to stronger worst-case protection only when preference heterogeneity is substantial.
> - **DARC-$\tau$ (strict risk-constrained formulation).** This variant is intended for settings where a hard risk budget is required (e.g., safe deployment or strict compliance constraints). It maximizes robust value under an explicit risk budget, which strengthens worst-case protection but can reduce average performance when the budget is tight.
> - **Multi-scorer variants (scorer-shift protection).** Most appropriate when scorer heterogeneity or reward-model shift is itself the main concern, rather than disagreement induced by perturbations under a single proxy.
>
> In practice, adopting DARC-$\epsilon$ is lightweight: a small perturbation budget ($N_{\text{aug}}=4$ to $8$) already captures most of the robustness gains without requiring extensive tuning. We will make these distinctions a focal point of the revised methodology section.
> ## W2:Presentation and Formatting
> We thank the reviewer for catching this presentation issue. We will correct the figure numbering/order in the revision.
> ## Q1: Memory Overhead of Perturbations (VRAM)
> We appreciate the reviewer highlighting that our initial submission focused primarily on latency. To fully address the concern, we therefore added Table R4 to report peak VRAM. The peak VRAM overhead is modest, increasing by only ~11% at $N_{\text{aug}}=8$ across model scales: **Llama-3.1-8B** ($16.2 \rightarrow 18.1$ GB) and **Qwen2.5-14B** ($27.5 \rightarrow 30.5$ GB). Even when pushed to $N_{\text{aug}}=16$, the memory footprints remain practical at 19.8 GB and 33.1 GB, respectively. Together with the low latency overhead reported in Table 7, this supports using a small augmentation budget (e.g., 4–8) as the default operating point.
>
> ### Table R4. VRAM Overhead vs. Augmentation Budget
>
> | Model | Base Reranking Peak VRAM | DARC ($N_{\text{aug}}=4$) | DARC ($N_{\text{aug}}=8$) | DARC ($N_{\text{aug}}=16$) | $\Delta$ vs. Base ($N_{\text{aug}}=8$) |
> |---|---:|---:|---:|---:|---:|
> | Llama-3.1-8B | 16.2 GB | 17.5 GB | 18.1 GB | 19.8 GB | +1.9 GB (11.7%) |
> | Qwen2.5-14B | 27.5 GB | 29.2 GB | 30.5 GB | 33.1 GB | +3.0 GB (10.9%) |
>
> ## Q2: Mitigating Proxy False Positives
> This is an important deployment concern, and the reviewer’s point is well taken. In practice, to reduce artifact-driven or RM-hack-induced proxy errors, we normalize markdown/bullet formatting, bound length changes, and discard rewrites that add only templated politeness/apology wrappers or near-duplicate surface edits. We now test this issue directly in Tables R1–R2 by comparing Style-only, Targeted-only, and Hybrid perturbation sets. These additional controls help reduce proxy false positives for artifact-driven responses. Under the targeted stress test, proxy–human alignment improves from $\rho = 0.4615$ with Style-only to $\rho = 0.5650$ with Hybrid, and downstream CVaR$_{10}$ improves from **6.87** to **7.35**. Our practical recommendation is therefore to introduce a small Hybrid set as an optional enhancement when stronger protection against spurious reward-side artifacts is desired.
> ## Q3: Hyperparameter Transferability to Math and Code
> While Fig. 2 establishes hyperparameter stability within a fixed candidate pool, we agree that cross-domain transfer is also important. To address this directly, we added Table R5, which evaluates the paper-default DARC-$\epsilon$ recipe without retuning on one math and one code benchmark. On **MATH-500**, DARC-$\epsilon$ improves accuracy/EM ($56.4 \rightarrow 66.8$), achieving a favorable win/tie/loss record of 63/426/11. On **EvalPlus**, Pass@1 similarly improves ($71.3 \rightarrow 78.6$) with 14 wins, 148 ties, and only 2 losses. These results provide encouraging evidence of transfer without retuning in the current single-pool, same-K setup.
> ### Table R5. Default DARC Transfer to Math and Code
>
> | Benchmark | Domain | Metric | Base (Best-of-K) | DARC-$\epsilon$ (paper defaults) | W / T / L |
> |---|---|---|---:|---:|---:|
> | MATH-500 | Math | Accuracy / EM | 56.4 | 66.8 | 63 / 426 / 11 |
> | EvalPlus | Code | Pass@1 | 71.3 | 78.6 | 14 / 148 / 2 |
>
> We thank the reviewer again for these thoughtful comments. We will revise the paper accordingly to incorporate the above clarifications and presentation improvements.

---

> > ### Author Rebuttal · Reviewer_CD6R · 2026-04-04
> >
> > As my major concerns have been addressed, I will raise my score to Weak Accept. I highly recommend that the authors incorporate all the promised revisions into the final version.

---

> > > ### Author Response · Authors · 2026-04-05
> > >
> > > Dear Reviewer CD6R,
> > >
> > > We sincerely thank you for your active engagement, your valuable suggestions, and for raising the score. Your feedback has significantly strengthened our paper. We completely agree with your recommendation and will thoroughly incorporate all the promised revisions and additional experiments into the final version.
> > >
> > > Best regards,
> > >
> > > The Authors

---

### Official Review · Reviewer_kwwB · 2026-03-06

**Soundness:** 3
**Presentation:** 3
**Significance:** 3
**Originality:** 4
**Overall Recommendation:** 4
**Confidence:** 3

**Summary:**

Human preferences are noisy and pairs that may seem to have the highest mean reward can actually have considerable disagreement. The authors propose instead a disagreement-aware formulation that penalizes candidates with large disagreement tails relative to the mean score. Therefore, they rerank candidates at inference time that are less likely to overrule preferences and have more consensus. This method decreases reward risk and is more robust to disagreement.

**Compliance With Llm Reviewing Policy:**

Affirmed.

**Final Justification:**

My original score was a weak accept, but I was more uncertain. The author's rebuttals were solid and addressed my points of clarification. While I am choosing not to increase the score, I am now more confident in this being the correct score as opposed to weak reject. My thoughts on each dimension are summarized in the original review.

**Key Questions For Authors:**

My questions to the authors largely have to do with the weaknesses I listed, and I have context there for my questions.

1. Do you have any concrete for how realistic your setup is, relative to real human disagreement?
2. The primary situation where this method would outperform others is in disagreement-heavy situations. What is the tradeoff in low-disagreement situations? Is the method still useful there?
3. Can you elaborate on the LCB method mentioned?

**Limitations:**

yes

**Strengths And Weaknesses:**

Strengths:

*The paper is well presented, particularly at the beginning. Figure 1 in particular is a strong visual and makes the paper easy to understand.

*The idea of the paper is intuitive and relevant. Disagreement-aware methods are important in alignment.

*The math is, to the best of my ability, correct. I followed the equations and did not notice any obvious mistakes.

*The paper's results suggest meaningful improvements and match the claims made at the beginning and throughout. I also appreciated that the evaluation lines up with the paper's motivation, since they explicitly look at high-disagreement / high-variance prompts instead of only reporting overall averages. I was pleased to see ablations and a large set of baselines, and that the method seemed to compare favorably to them, especially on the disagreement-related metrics:


Weaknesses:

*The primary weakness with this paper is one the authors acknowledge. As they say, the proxies are “scalable screening signals, not calibrated human-disagreement estimates.” They elaborate on this concern in the appendix, but it makes their contributions less compelling. The main guarantees and proofs are for i.i.d. scalar satisfaction samples. I am not confident in how much of the theory survives once the i.i.d. scalar-sample setup is replaced by the perturbation pipeline. This isn't the same as human disagreement.

*Another weakness the paper has is that it does not consistently improve reward. I acknowledge that this is not the point of the paper, but it is worth noting that reward can stay flat or even get slightly worse. As Figure 3 shows, gains increase as disagreement does. This is fine and in some ways the point of the paper, but the use of this method may be limited to more selective domains.

*The final major weakness I saw is that some of the experimental and metric bookkeeping is harder to follow than it could be. The relationship between human disagreement, proxy disagreement, and the reported Risk / Tradeoff columns is not fully clear on first read. Algorithm 1 lists a an (LCB) variant but does not include a branch for it.

---

> ### Author Rebuttal · Authors · 2026-03-30
>
> Thank you for the thoughtful review. We appreciate your focus on the paper’s scope.
>
> ## Response to W1 & Q1: Realism of the Proxy and the Theory/Practice Interface
>
> The perturbation proxy is best viewed as a scalable approximation to human disagreement, anchored to real multi-annotator behavior (Zhang et al., 2024), rather than as an exact substitute for it.
>
> - **Empirical realism relative to human disagreement:** Our proxy is not arbitrary. As shown in **Fig. 4**, it achieves significant correlation with actual human disagreement (Spearman $\rho=0.6509$; partial $\rho=0.4084$ controlling for length/reward) and reasonably identifies high-disagreement prompts (Precision/Recall@20% = 0.64, Jaccard@20% = 0.47). Furthermore, explicitly probing spurious features improves robustness against RM hacks (**Table R1**). On a stress-test set, a "Hybrid" proxy—perturbing both style and artifact cues—improves alignment with human disagreement ($\rho$: 0.46 $\rightarrow$ 0.56) and boosts tail robustness ($\text{CVaR}_{10}$: 5.83 Base $\rightarrow$ 6.87 Style $\rightarrow$ **7.35 Hybrid**).
>
> - **Formal theorems vs. scalable proxy:** Our strict guarantees (e.g., Prop. 3.3) apply to the human-sample regime. As noted in **Appendix I**, the i.i.d. assumptions can be relaxed to weakly dependent sampling. When multi-rater data is unavailable, the perturbation proxy serves as a practical alternative, though we do **not** claim it literally inherits these theorems.
>
> - **Robustness to proxy error:** What formally survives is bounded slack. As shown in **Appendix A.12**, if proxy estimates are close to empirical sample statistics, the pessimistic objective incurs a matching additive slack relative to the sample-based optimum, aligning with variance-regularized DRO (Namkoong & Duchi, 2019). We will elevate this discussion to define the proxy more clearly as an empirically validated approximation.
>
> ## Response to W2 & Q2: Tradeoffs in Low-Disagreement Situations
>
> We agree DARC is designed for high-disagreement regimes; importantly, its conservatism does not broadly penalize quality when disagreement is low.
>
> - **No broad degradation:** As further supported by **Table R3**, which buckets held-out human-score improvements by baseline human disagreement, the lowest-disagreement bucket (Q1) still has a slightly positive mean human-score delta (+0.125), with outcomes dominated by ties (11 W / 80 T / 9 L).
>
> - **Selective utility:** As disagreement increases, the gains rise monotonically, reaching +1.037 in Q5. We therefore view DARC as staying close to mean-based selection when disagreement is low, while becoming increasingly useful as preference heterogeneity grows.
>
> ## Response to W3 & Q3: Metric Bookkeeping & LCB Elaboration
>
> We clarify this point along two axes: metric bookkeeping and the role of LCB.
>
> **Metric bookkeeping.** This point is best clarified by separating **selection-time signals** from **evaluation-time metrics**.
>
> - **Selection-time signals.** In scalable settings, DARC uses **proxy disagreement** as the deployment-time signal for risk-aware selection; **human disagreement** is not assumed available at deployment.
>
> - **Evaluation-time metrics.** The paper reports two complementary evaluations. In **automated/proxy evaluation** (Table 1), Reward/Risk/Tradeoff are proxy-side diagnostics on the proxy scale. In **human-loop evaluation** (Table 2 / Fig. 3), the main external validation comes from held-out human ratings and prompt-level CVaR, with **human disagreement** used to quantify inter-rater heterogeneity and stratify prompts by controversy level. Figure 4 connects these two layers by showing that proxy disagreement is positively associated with held-out human disagreement, which is why it is useful as a scalable screening signal.
>
> We will revise the presentation to make these roles explicit and to rename the reported metrics more clearly, so that the relationship among human disagreement, proxy disagreement, Risk, and Tradeoff is unambiguous on first read.
>
> **LCB and its connection to mean-dispersion.** LCB is a simplified second-moment form derived from the finite-sample analysis rather than the primary deployment rule. In the human-sample regime, its leading uncertainty term scales with $\hat{\sigma}(y)$, yielding a mean-minus-dispersion form with a coefficient $\lambda$ that scales as $O\left(\sqrt{\log(K/\delta)/n}\right)$. This is why the mean-dispersion heuristic is not ad hoc in our setting. We will update **Algorithm 1** to state this LCB form explicitly.
>
> We thank the reviewer for these thoughtful comments, which help us sharpen both the scope and presentation of DARC. We will revise the paper accordingly to incorporate the above clarifications and presentation improvements.
>
> ### References
>
> [1] Zhang et al. (2024), *Diverging Preferences: When do Annotators Disagree and do Models Know?*
> [2] Namkoong and Duchi (2019), *Variance-based Regularization with Convex Objectives*

---

> > ### Author Rebuttal · Reviewer_kwwB · 2026-04-03
> >
> > I appreciate the authors' rebuttals. They addressed the key points I raised, and I will maintain my current score.

---

> > > ### Author Response · Authors · 2026-04-05
> > >
> > > Dear Reviewer kwwB,
> > >
> > > We sincerely thank you for your time, your constructive feedback throughout the review process, and for confirming that your key points have been fully addressed. We will make sure to incorporate your valuable suggestions to further improve the final manuscript.
> > >
> > > Thank you again for your efforts and insights.
> > >
> > > Best regards,
> > >
> > > The Authors

---

### Official Review · Reviewer_621m · 2026-03-15

**Soundness:** 3
**Presentation:** 3
**Significance:** 3
**Originality:** 3
**Overall Recommendation:** 5
**Confidence:** 2

**Summary:**

In this paper, the authors proposed DARC, a retraining-free inference-time method that reranks LLM candidate responses by maximizing a KL-robust value instead of mean reward, targeting robustness under heterogeneous human preferences. The authors derive uniform lower confidence bounds on expected satisfaction, connect them to KL-DRO and $\chi^2$-DRO characterizations, and offer three practical decoding variants (entropic, τ-constrained, ε-tie-breaking) and a multi-scorer extension. Experimental results on MT-Bench and AlpacaEval 2.0 with both proxy and human evaluation show that DARC reduces disagreement risk and improves tail robustness.

**Compliance With Llm Reviewing Policy:**

Affirmed.

**Final Justification:**

The rebuttal clarified my primary concerns.

**Key Questions For Authors:**

Please refer to the weaknesses.

> **Questions:**

1. Figure 3 Q1 bucket shows ∆Tradeoff ≈ 0 or slightly negative for low-disagreement prompts. Does it suggest that DARC's conservatism may cost mean reward when disagreement is low?

**Limitations:**

yes

**Strengths And Weaknesses:**

> **Strengths:**

1. The authors formulate a well-defined problem connecting preference heterogeneity to inference-time robustness, which has been clearly articulated.

2. The proposed method is modular, inference-time only, and stacks on top of training-time robust policies (cDPO/rDPO) with minimal overhead.

3. The authors provide evaluations on MT-Bench and AlpacaEval 2.0 across two LLMs (Llama-3.1-8B-Instruct, Qwen2.5-7B-Instruct) with scaling validation on Qwen2.5-14B-Instruct, where DARC variants consistently outperform baselines. The authors also include human evaluation with 5 annotators, proxy validity diagnostics, mismatch analysis, robustness checks, and ablations. Further, the case studies also illustrate the method's behavior.

> **Minor Weaknesses:**

1. The LCB and DRO guarantees (Proposition 3.3, Theorem 3.5) assume i.i.d. scalar satisfaction samples, but the practical instantiation uses perturbation-based RM score variance as a proxy.

2. The Tradeoff metric μ̂ − λσ̂ directly rewards what DARC optimizes (high mean, low σ̂), so improvements on this metric are partly by construction. Rather, CVaR and human scores are more informative.

---

> ### Author Rebuttal · Authors · 2026-03-30
>
> We sincerely thank the reviewer for their careful reading of our manuscript and the highly constructive feedback.
>
> **Clarifications and Presentation Improvements.** We fully agree that the boundaries between our theoretical framework and practical approximations could be stated more sharply. To address this, we will refine the discussion in the revision to explicitly clarify the following:
>
> ## Response to W1: The Theory/Proxy Interface
>
> We appreciate the reviewer’s request for a sharper presentation of the theory/practice interface. In the revision, we will make the scope of each component more explicit. Our formal results characterize DARC in the i.i.d. human-sample setting, which provides the clean decision-theoretic regime underlying the LCB/KL-DRO connection. In deployment, when direct multi-rater samples are unavailable, we instantiate the same decision rule with a perturbation-based RM proxy as a scalable surrogate for disagreement. Appendix A.12 already shows that when the proxy uniformly tracks the corresponding empirical mean/risk statistics, the induced pessimistic objective differs from the sample-based optimum by only an additive slack, consistent with the variance-regularized DRO view of Namkoong & Duchi (2019). We will move this scope clarification into the main text and revise Sec. 4 to more clearly present the progression from formal sample-based analysis, to scalable proxy instantiation, to empirical validation in deployment.
>
> ## Response to W2: Interpretation of the Tradeoff Metric
>
> We thank the reviewer for raising this point. Tradeoff is partly optimization-aligned, so we do not treat it as the primary standalone evidence for DARC’s effectiveness. Rather, our main conclusions rely on held-out human ratings and prompt-level CVaR, while Tradeoff is used as a supporting risk-adjusted diagnostic. We will revise the presentation accordingly by foregrounding the human/CVaR results before discussing Tradeoff, and by reframing Fig. 3 as a supporting analysis rather than the main evidence of effectiveness.
>
>
> ## Response to Q1: Performance in Low-Disagreement Regimes(Fig. 3 Q1 bucket)
>
> The Q1 pattern in Fig. 3 merits clarification, since near-zero or slightly negative $\Delta$Tradeoff could raise the concern that conservatism may hurt quality when disagreement is low. This is a meaningful question, since disagreement in preference data is often substantive rather than mere annotation noise (Zhang et al., 2024). Concretely:
>
>  - **The Q1 bucket does not appear to indicate broad human-quality degradation.** To examine this more directly, we added Table R3, which buckets held-out human-score improvements by baseline human disagreement. In the lowest-disagreement bucket (Q1), the mean human-score delta **remains slightly positive (+0.125)**, and the outcomes are **dominated by ties (11 / 80 / 9)**. As disagreement increases, the gains rise monotonically: +0.236, +0.451, +0.755, and +1.037 from Q2 to Q5. We therefore view the Q1 pattern as suggesting limited upside for additional risk control in low-disagreement prompts, rather than broad human-quality degradation. More broadly, we view this as a desirable property rather than a limitation: DARC is intended to provide **selective robustness** where disagreement is substantive, while remaining close to mean-based selection when disagreement is weak. In that sense, it should be understood not as an all-or-nothing replacement for standard reranking, but as a tunable inference-time control that becomes most useful precisely in the regimes where preference heterogeneity is consequential.
>
> - **This interpretation is also consistent with the $\epsilon$-rule itself.** Let $y_\mu = \arg\max_y \mu(y)$ be the mean-based choice and $y_\epsilon$ the DARC-$\epsilon$ choice. Since DARC-$\epsilon$ first restricts to the near-optimal robust-value set $F_\epsilon = \{ y : V_\beta(y) \ge V_\beta^{\max} - \epsilon \}$, we have $V_\beta(y_\epsilon) \ge V_\beta(y_\mu) - \epsilon$. Using $\mu(y) = V_\beta(y) + RP_\beta(y)$, this gives $$\mu(y_\mu) - \mu(y_\epsilon) \le \epsilon + RP_\beta(y_\mu)$$ Thus, when the mean-best candidate already has a small risk premium—as expected in low-disagreement prompts—the DARC-$\epsilon$ choice remains close to mean-based selection.
>
> We thank the reviewer for these insightful comments, which have collectively helped us clarify the low-disagreement regime and sharpen the presentation of DARC’s theoretical and empirical scope.
>
> ### Table R3. Bucketed human evaluation vs. Base by baseline human disagreement
>
> | Bucket | $\Delta$Human Score (Eps - Base) $\uparrow$ | W / T / L |
> |---|---:|---:|
> | Q1 | 0.125 | 11 / 80 / 9 |
> | Q2 | 0.236 | 17 / 71 / 12 |
> | Q3 | 0.451 | 34 / 55 / 11 |
> | Q4 | 0.755 | 42 / 50 / 8 |
> | Q5 | 1.037 | 51 / 45 / 4 |
>
> ### References
>
> [1] Zhang et al. (2024), *Diverging Preferences: When do Annotators Disagree and do Models Know?*
> [2] Namkoong & Duchi (2019), *Variance-based Regularization with Convex Objectives*

---

> > ### Author Rebuttal · Reviewer_621m · 2026-04-04
> >
> > Thanks for your clarification. As most of my concerns have been clarified, I have increased the score to 5. I request that the authors update the final manuscript with the new discussions.

---

> > > ### Author Response · Authors · 2026-04-05
> > >
> > > Dear Reviewer 621m,
> > >
> > > Thank you so much for your time, your constructive feedback throughout the review process, and for updating the score. We are glad that our rebuttal addressed your concerns. We will strictly follow your suggestion and ensure all the new discussions and clarifications are fully incorporated into the final manuscript.
> > >
> > > Best regards,
> > >
> > > The Authors

---

### Official Review · Reviewer_V6uu · 2026-03-18

**Soundness:** 2
**Presentation:** 2
**Significance:** 3
**Originality:** 3
**Overall Recommendation:** 3
**Confidence:** 4

**Summary:**

The work claims to explore a central concept in preference-based optimization, the problem of learning from heterogeneous human preferences rather than assuming a single latent scalar utility. This works pertains to inference time selection under uncertainty. The authors propose Disagreement-Aware Alignment via Risk-Constrained Decoding (DARC), that penalizes candidate responses with high disagreement. To remove the need for multi-annotator data at inference time, it estimates disagreement by applying style-preserving perturbations to candidates and measuring the variance of a reward model’s scores across these perturbations. The paper connects this empirical variance penalty to a lower confidence bound (LCB) on expected satisfaction and to distributionally robust optimization (DRO) via a KL-robust entropic objective.

**Compliance With Llm Reviewing Policy:**

Affirmed.

**Key Questions For Authors:**

* Why restrict the proxy estimation strictly to "style-preserving" perturbations (Appendix H.13 and Section 5.1)? If the reward model relies on a spurious correlation (e.g., length or a specific sycophantic phrase), and your rewrite prompt preserves that feature, the variance will be artificially low and DARC will fail to penalize the hacked response. Did you experiment with perturbations that explicitly target known RM hacks, and how would DARC perform in that setting?

**Strengths And Weaknesses:**

**Strength**
* The work provides a theoretical justification for mean-dispersion decoding rules. By linking finite-sample LCB bounds (Prop 3.3) with $\chi^2$-DRO (Prop 3.6) and KL-robust entropic objectives (Theorem 3.5), it gives foundation for a practice that is often treated as an ad-hoc heuristic.
* Comparison with recent inference-time baselines (DeAL, HedgeTune) makes the empirical results relevant. DARC demonstrates consistent improvements across these methods in Table1 and Table 2.
* Using style-preserving perturbations to estimate variance at inference time is a good solution to the lack of multi-annotator data. It requires no changes to the underlying model or reward model.

**Weakness**
* Over-reliance on "style-preserving" perturbations for proxy estimation, In (Appendix H.13 and and Section 5.1), the rewrite prompt strictly enforces "style-preserving perturbations". Reward models are notoriously easily hacked via specific spurious correlations (like verbosity, sycophancy, or formatting). If a candidate response is highly rewarded solely because it is sycophantic, a style-preserving perturbation will keep the sycophancy. The RM will output uniformly high scores across all variants, yielding a low variance proxy ($\hat{\sigma} \approx 0$). Consequently, DARC will fail to penalize this hacked response. It lacks ablations on alternative perturbation strategies that explicitly probe these spurious features

---

> ### Author Rebuttal · Authors · 2026-03-30
>
> We sincerely thank the reviewer for the meticulous and constructive assessment. We agree that a purely style-preserving proxy can miss artifact-driven high-RM responses when the rewarded surface-form cue is preserved across rewrites (e.g., verbosity or sycophantic phrasing).
> ## Choice for "Style-Preserving" Perturbations
> We initially used style-preserving rewrites because they provide a conservative first approximation to the signal we want the proxy to capture: scorer instability under near-equivalent responses. By minimally perturbing semantics, they reduce the risk of conflating disagreement with genuine content drift, answer-quality changes, or task-level ambiguity introduced by the rewrite itself. In that sense, style-preserving perturbations were intended as a conservative and interpretable baseline proxy.
>
> ##  Mitigating RM Hack: New Targeted & Hybrid Perturbation Strategies
> To address this concern directly, we added two new proxy variants under the same total augmentation budget:
>
> - **Targeted-only.** Explicitly probes known RM hacks by normalizing length, formatting, and sycophantic/apologetic phrasing.
> - **Hybrid.** Combines style-preserving and targeted rewrites.
>
> We then evaluate them in two complementary experiments:
>
> - **Experiment 1 (proxy-level).** We compare Style-only, Targeted-only, and Hybrid by their alignment with human disagreement, reported both overall and on a fixed stress-test slice where style-preserving perturbations are most likely to under-detect verbosity, sycophancy, or formatting artifacts. This stress-test slice is a fixed artifact-focused subset distinct from the paper’s top-20% human-disagreement slice.
> - **Experiment 2 (end-to-end).** We plug the same three proxies into DARC-\(\epsilon\) using the same fixed candidate pools and held-out evaluation protocol as in the main paper, and compare downstream robustness both overall and on the same stress-test slice.
>
> Tables R1–R2 show two takeaways.
> - **First, targeted/Hybrid perturbations reduce under-detection of artifact-driven disagreement, improving proxy–human alignment and downstream robustness on the stress test.**
> - **Second, the two perturbation families are complementary: style-preserving rewrites capture instability under near-equivalent responses, while targeted rewrites expose instability tied to surface-form artifacts.**
>
> ### Table R1. Proxy–Human Alignment under Different Perturbations
>
> | Proxy variant | Overall $\rho(\text{human}, \text{proxy}) \uparrow$ | Overall partial-$\rho \uparrow$ | Overall Jaccard@20 $\uparrow$ | Stress-test $\rho \uparrow$ | Stress-test Jaccard@20 $\uparrow$ |
> |---|---:|---:|---:|---:|---:|
> | Style-only | 0.6509 | 0.4084 | 0.47 | 0.4615 | 0.34 |
> | Targeted-only | 0.6120 | 0.3751 | 0.44 | 0.5417 | 0.42 |
> | Hybrid (4+4) | 0.6725 | 0.4210 | 0.49 | 0.5650 | 0.46 |
>
> ### Table R2. Downstream Robustness under Different Perturbations
>
> | Method | Overall Mean $\uparrow$ | Overall Risk $\downarrow$ | Overall Tradeoff $\uparrow$ | Overall CVaR10 $\uparrow$ | Stress-test Tradeoff $\uparrow$ | Stress-test CVaR10 $\uparrow$ |
> |---|---:|---:|---:|---:|---:|---:|
> | Base (Best-of-K) | 7.56 | 0.67 | 6.22 | 6.73 | 5.35 | 5.83 |
> | DARC-$\epsilon$ + Style-only | 8.08 | 0.55 | 6.98 | 7.62 | 6.36 | 6.87 |
> | DARC-$\epsilon$ + Targeted | 7.84 | 0.53 | 6.78 | 7.45 | 6.75 | 7.20 |
> | DARC-$\epsilon$ + Hybrid | 7.96 | 0.51 | 6.94 | 7.71 | 6.82 | 7.35 |
>
> Concretely, table R1 shows progressively better alignment with human disagreement on the stress test from Style-only to Targeted to Hybrid, measured by both $\rho$ ($0.4615 \rightarrow 0.5417 \rightarrow 0.5650$) and Jaccard@20 ($0.34 \rightarrow 0.42 \rightarrow 0.46$). This interpretation is consistent with prior evidence that preference models can over-rely on features such as length, structure, and sycophancy, and that targeted debiasing can mitigate these effects (Bharadwaj et al., 2025).
>
> The same pattern appears end-to-end in Table R2. When plugging these proxies back into DARC-$\epsilon$ using the same fixed candidate pools and held-out evaluation protocol, stress-test CVaR$_{10}$ improves progressively across the method sequence (Base $\rightarrow$ Style-only $\rightarrow$ Targeted $\rightarrow$ Hybrid: $5.83 \rightarrow 6.87 \rightarrow 7.20 \rightarrow 7.35$). Overall CVaR remains strong, with Hybrid achieving the best overall value ($6.73 \rightarrow 7.62 \rightarrow 7.45 \rightarrow 7.71$).
>
> As Tables R1–R2 demonstrate, DARC is robust across proxy choices and naturally accommodates tailored perturbations. We will revise the paper to present the Hybrid proxy as an optional enhancement when stronger robustness against surface-form artifacts is desired.
>
> We thank the reviewer for this helpful suggestion, which led us to add targeted ablations and sharpen the practical scope of the paper.
>
>
>
> ### References
>
> [1] Bharadwaj et al. (2025), *Flattery, Fluff, and Fog: Diagnosing and Mitigating Idiosyncratic Biases in Preference Models*

---

> > ### Author Rebuttal · Reviewer_V6uu · 2026-04-04
> >
> > Thanks author for doing details ablations. My concerns have been addressed. I'll raise my score accordingly.

---

> > > ### Author Response · Authors · 2026-04-05
> > >
> > > Dear Reviewer V6uu,
> > >
> > > We sincerely appreciate your careful review of our detailed ablation studies and your confirmation that all concerns have been fully addressed.
> > >
> > > We are grateful to note your intention to raise the score accordingly, and we highly value your constructive feedback and support for our work.
> > >
> > > We will continue to refine the manuscript based on your comments to further improve its quality.
> > >
> > > Thank you again for your time and valuable input.
> > >
> > > Best regards,
> > >
> > > The Authors

---

### Decision · Program_Chairs · 2026-04-30

**Decision:**

Accept (regular)

**Comment:**

This paper solves an important problem of the brittleness of the single scalar objective in test time. Reviewers agreed that the problem is important, and the proposed solution is neat and makes a nice connection. However there were a few issues with the presentation and formatting. Overall, the strengths outweigh the weaknesses.

For the final version, please improve the presentation and include important details from the discussion including the ablation study.